# Online Adaptive Policy Selection in Time-Varying Systems: No-Regret via Contractive Perturbations

**Yiheng Lin**
California Institute of Technology
Pasadena, CA, USA
`yihengl@caltech.edu`

**James A. Preiss**
California Institute of Technology
Pasadena, CA, USA
`japreiss@caltech.edu`

**Emile Anand**
California Institute of Technology
Pasadena, CA, USA
`eanand@caltech.edu`

**Yingying Li**
University of Illinois Urbana-Champaign
Urbana, IL, USA
`yl101@illinois.edu`

**Yisong Yue**
California Institute of Technology
Pasadena, CA, USA
`yyue@caltech.edu`

**Adam Wierman**
California Institute of Technology
Pasadena, CA, USA
`adamw@caltech.edu`

## Abstract

We study online adaptive policy selection in systems with time-varying costs and dynamics. We develop the Gradient-based Adaptive Policy Selection (GAPS) algorithm together with a general analytical framework for online policy selection via online optimization. Under our proposed notion of contractive policy classes, we show that GAPS approximates the behavior of an ideal online gradient descent algorithm on the policy parameters while requiring less information and computation. When convexity holds, our algorithm is the first to achieve optimal policy regret. When convexity does not hold, we provide the first local regret bound for online policy selection. Our numerical experiments show that GAPS can adapt to changing environments more quickly than existing benchmarks.

## 1 Introduction

We study the problem of online adaptive policy selection for nonlinear time-varying discrete-time dynamical systems. The dynamics are given by $x_{t+1} = g_t(x_t, u_t)$, where $x_t$ is the state and $u_t$ is the control input at time $t$. The policy class is a time-varying mapping $\pi_t$ from the state $x_t$ and a policy parameter $\theta_t$ to a control input $u_t$. At every time step $t$, the online policy incurs a stage cost $c_t = f_t(x_t, u_t)$ that depends on the current state and control input. The goal of policy selection is to pick the parameter $\theta_t$ online to minimize the total stage costs over a finite horizon $T$.

Online adaptive policy selection and general online control have received significant attention recently [1]–[8] because many control tasks require running the policy on a single trajectory, as opposed to restarting the episode to evaluate a different policy from the same initial state. Adaptivity is also important when the dynamics and cost functions are time-varying. For example, in robotics, time-varying dynamics arise when we control an aircraft under changing wind conditions [9].

---

[†]This work is supported by NSF Grants CNS-2146814, CPS-2136197, CNS-2106403, NGSDI-2105648, CCF-1918865, and Gift from Latitude AI, with additional support for Yiheng Lin provided by Amazon AI4Science Fellowship and PIMCO Graduate Fellowship in Data Science.

37th Conference on Neural Information Processing Systems (NeurIPS 2023).

In this paper, we are interested in developing a unified framework that can leverage a broad suite of theoretical results from online optimization and efficiently translate them to online policy selection, where efficiency includes both preserving the tightness of the guarantees and computational considerations. A central issue is that, in online policy selection, the stage cost $c_t$ depends on all previously selected parameters $(\theta_0, \ldots, \theta_{t-1})$ via the state $x_t$. Many prior works along this direction have addressed this issue by finite-memory reductions. This approach led to the first regret bound on online policy selection, but the bounds are not tight, the computational cost can be large, and the dynamics and policy classes studied are restrictive [1, 3, 6-8].

**Contributions.** We propose and analyze the algorithm Gradient-based Adaptive Policy Selection (GAPS, Algorithm 1) to address three limitations of existing results on online policy selection. First, under the assumption that $c_t$ is a convex function of $(\theta_0, \ldots, \theta_t)$, prior work left a $\log T$ regret gap between OCO and online policy selection. We close this gap by showing that GAPS achieves the optimal regret of $O(\sqrt{T})$ (Theorem 3.3) [1]. Second, many previous approaches require oracle access to the dynamics/costs and expensive resimulation from imaginary previous states. In contrast, GAPS only requires partial derivatives of the dynamics and costs along the visited trajectory, and computes $O(\log T)$ matrix multiplications at each step. Third, the application of existing regret analysis frameworks is limited to specific policy classes and systems because they require $c_t$ to be convex in $(\theta_0, \ldots, \theta_t)$. We address this limitation by showing the first local regret bound for online policy selection when the convexity does not hold. Specifically, GAPS achieves the local regret of $O(\sqrt{(1+V)T})$, where $V$ is a measure of how much $(g_t, f_t, \pi_t)$ changes over the entire horizon.

To derive these performance guarantees, we develop a novel proof framework based on a general exponentially decaying, or "contractive", perturbation property (Definition 2.6) on the policy-induced closed-loop dynamics. This generalizes a key property of disturbance-action controllers [e.g. 1, 8] and includes other important policy classes such as model predictive control (MPC) [e.g. 11] and linear feedback controllers [e.g. 12]. Under this property, we prove an approximation error bound (Theorem 3.2), which shows that GAPS can mimic the update of an ideal online gradient descent (OGD) algorithm [13] that has oracle knowledge of how the current policy parameter $\theta_t$ would have performed if used exclusively over the whole trajectory. This error bound bridges online policy selection and online optimization, which means regret guarantees on OGD for online optimization can be transferred to GAPS for online policy selection.

In numerical experiments, we demonstrate that GAPS can adapt faster than an existing follow-the-leader-type baseline in MPC with imperfect disturbance predictions, and outperforms a strong optimal control baseline in a nonlinear system with non-i.i.d. disturbances. The source code for all experiments is published at `https://www.github.com/jpreiss/adaptive_policy_selection`.

**Related Work.** Our work is related to online control and adaptive-learning-based control [2, 14-19], especially online control with adversarial disturbances and regret guarantees [1, 6-8, 20, 21]. For example, there is a rich literature on policy regret bounds for time-invariant dynamics [1, 6, 8, 15, 20, 22]. There is also a growing interest in algorithms for time-varying systems with small adaptive regret [7, 21], dynamic regret [23-25], and competitive ratio [26-29]. Many prior works study a specific policy class called disturbance-action controller (DAC) [1, 3, 6-8]. When applied to linear dynamics $g_t$ with convex cost functions $f_t$, DAC renders the stage cost $c_t$ a convex function in past policy parameters $(\theta_0, \ldots, \theta_t)$. Our work contributes to the literature by proposing a general contractive perturbation property that includes DAC as a special case, and showing local regret bounds that do not require $c_t$ to be convex in $(\theta_0, \ldots, \theta_t)$. A recent work also handles nonconvex $c_t$, but it studies an episodic setting and requires $c_t$ to be "nearly convex", which holds under its policy class [30].

In addition to online control, this work is also related to online learning/optimization [13, 31, 32], especially online optimization with memory and/or switching costs, where the cost at each time step depends on past decisions. Specifically, our online adaptive policy selection problem is related to online optimization with memory [29, 33-39]. Our analysis for GAPS provides insight on how to handle indefinite memory when the impact of a past decision decays exponentially with time.

Our contractive perturbation property and the analytical framework based on this property are closely related to prior works on discrete-time incremental stability and contraction theory in nonlinear systems [40-46], as well as works that leverage such properties to derive guarantees for (online) controllers [47-49]. In complicated systems, it may be hard to design policies that provably satisfy

---

[1]A concurrent work [10] also closes this gap in a less general setting. See Section 3.2 for a discussion.

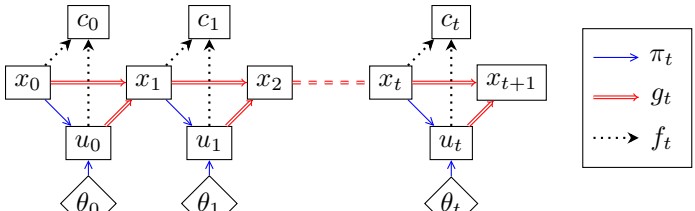

Figure 1: Diagram of the causal relationships between states, policy parameters, control inputs, and costs.

these properties. This motivates some recent works to study neural-based approaches that can learn a controller together with its certificate for contraction properties simultaneously [50, 51]. Our work contributes to this field by showing that, when the system satisfies the contractive perturbation property, one can leverage this property to bridge online policy selection with online optimization.

**Notation.** We use $[t_1 : t_2]$ to denote the sequence $(t_1, \ldots, t_2)$, $a_{t_1:t_2}$ to denote $(a_{t_1}, a_{t_1+1}, \ldots, a_{t_2})$ for $t_1 \leq t_2$, and $a_{\times \tau}$ for $(a, \ldots, a)$ with $a$ repeated $\tau \geq 0$ times. We define $q(x, Q) = x^\top Q x$. Symbols $\mathbf{1}$ and $\mathbf{0}$ denote the all-one and all-zero vectors/matrices respectively, with dimension implied by context. The Euclidean ball with center $\mathbf{0}$ and radius $R$ in $\mathbb{R}^n$ is denoted by $B_n(0, R)$. We let $\|\cdot\|$ denote the (induced) Euclidean norm for vectors (matrices). The diameter of a set $\Theta$ is $\mathsf{diam}(\Theta) := \sup_{x,y \in \Theta} \|x - y\|$. The projection onto the set $\Theta$ is $\Pi_\Theta(x) = \arg\min_{y \in \Theta} \|y - x\|$.

## 2 Preliminaries

We consider online policy selection on a single trajectory. The setting is a discrete-time dynamical system with state $x_t \in \mathbb{R}^n$ for time index $t \in \mathcal{T} := [0 : T - 1]$. At time step $t \in \mathcal{T}$, the policy picks a control action $u_t \in \mathbb{R}^m$, and the next state and the incurred cost are given by:

$$\text{Dynamics: } x_{t+1} = g_t(x_t, u_t), \qquad \text{Cost: } c_t := f_t(x_t, u_t),$$

respectively, where $g_t(\cdot, \cdot)$ is a time-varying dynamics function and $f_t(\cdot, \cdot)$ is a time-varying stage cost. The goal is to minimize the total cost $\sum_{t=0}^{T-1} c_t$.

We consider parameterized time-varying policies of the form of $u_t = \pi_t(x_t, \theta_t)$, where $x_t$ is the current state at time step $t$ and $\theta_t \in \Theta$ is the current policy parameter. $\Theta$ is a closed convex subset of $\mathbb{R}^d$. We assume the dynamics, cost, and policy functions $\{g_t, f_t, \pi_t\}_{t \in \mathcal{T}}$ are oblivious, meaning they are fixed before the game begins. The online policy selection algorithm optimizes the total cost by selecting $\theta_t$ sequentially. We illustrate how the policy parameter sequence $\theta_{0:T-1}$ affects the trajectory $\{x_t, u_t\}_{t \in \mathcal{T}}$ and per-step costs $c_{0:T-1}$ in Figure 1. The online algorithm has access to the partial derivatives of the dynamics $f_t$ and cost $g_t$ *along the visited trajectory*, but does not have oracle access to the $f_t, g_t$ for arbitrary states and actions.

We provide two motivating examples for our setting. Appendix H contains more details and a third example. The first example is MPC with confidence coefficients, a generalization of [38].

**Example 2.1** (MPC with Confidence Coefficients). *Consider a linear time-varying (LTV) system* $g_t(x_t, u_t) = A_t x_t + B_t u_t + w_t$, *with time-varying costs* $f_t(x_t, u_t) = q(x_t, Q_t) + q(u_t, R_t)$. *At time $t$, the policy observes* $\{A_{t:t+k-1}, B_{t:t+k-1}, Q_{t:t+k-1}, R_{t:t+k-1}, w_{t:t+k-1|t}\}$, *where $w_{\tau|t}$ is a (noisy) prediction of the future disturbance $w_\tau$. Then, $\pi_t(x_t, \theta_t)$ commits the first entry of*

$$\arg\min_{u_{t:t+k-1|t}} \sum_{\tau=t}^{t+k-1} f_\tau(x_{\tau|t}, u_{\tau|t}) + q(x_{t+k|t}, \tilde{Q}) \tag{1}$$

$$\text{s. t. } x_{t|t} = x_t, \quad x_{\tau+1|t} = A_\tau x_{\tau|t} + B_\tau u_{\tau|t} + \lambda_t^{[\tau-t]} w_{\tau|t} : t \leq \tau < t+k,$$

*where* $\theta_t = \left(\lambda_t^{[0]}, \lambda_t^{[1]}, \ldots, \lambda_t^{[k-1]}\right)$, $\Theta = [0, 1]^k$ *and $\tilde{Q}$ is a fixed positive-definite matrix. Intuitively,* $\lambda_t^{[i]}$ *represents our level of confidence in the disturbance prediction $i$ steps into the future at time step $t$, with entry $1$ being fully confident and $0$ being not confident at all.*

The second example studies a nonlinear control model motivated by [12, 47].

**Example 2.2** (Linear Feedback Control in Nonlinear Systems). *Consider a time-varying nonlinear control problem with dynamics* $g_t(x_t, u_t) = A x_t + B u_t + \delta_t(x_t, u_t)$ *and costs* $f_t(x_t, u_t) = q(x_t, Q) +$

$q(u_t, R)$. Here, the nonlinear residual $\delta_t$ comes from linearization and is assumed to be sufficiently small and Lipschitz. Inspired by [12], we construct an online policy based on the optimal controller $u_t = -\bar{K}x_t$ for the linear-quadratic regulator $\mathrm{LQR}(A, B, Q, R)$. Specifically, we let $\pi_t(x_t, \theta_t) = -K(\theta_t)x_t$ where $K$ is a mapping from $\Theta$ to $\mathbb{R}^{n \times m}$ such that $\|K(\theta_t) - \bar{K}\|$ is uniformly bounded.

## 2.1 Policy Class and Performance Metrics

In our setting, the state $x_t$ at time $t$ is uniquely determined by the combination of 1) a state $x_\tau$ at a previous time $\tau < t$, and 2) the parameter sequence $\theta_{\tau:t-1}$. Similarly, the cost at time $t$ is uniquely determined by $x_\tau$ and $\theta_{\tau:t}$. Since we use these properties often, we introduce the following notation.

**Definition 2.3** (Multi-Step Dynamics and Cost). *The multi-step dynamics $g_{t|\tau}$ between two time steps $\tau \leq t$ specifies the state $x_t$ as a function of the previous state $x_\tau$ and previous policy parameters $\theta_{\tau:t-1}$. It is defined recursively, with the base case $g_{\tau|\tau}(x_\tau) := x_\tau$ and the recursive case*

$$g_{t+1|\tau}(x_\tau, \theta_{\tau:t}) = g_t(z_t, \pi_t(z_t, \theta_t)), \ \forall\, t \geq \tau,$$

*in which $z_t := g_{t|\tau}(x_\tau, \theta_{\tau:t-1})$.[2] The multi-step cost $f_{t|\tau}$ specifies the cost $c_t$ as function of $x_\tau$ and $\theta_{\tau:t}$. It is defined as $f_{t|\tau}(x_\tau, \theta_{\tau:t}) := f_t(z_t, \pi_t(z_t, \theta_t))$.*

In this paper, we frequently compare the trajectory of our algorithm against the trajectory that would arise from applying a fixed parameter $\theta$ since time step 0, which we denote as $\hat{x}_t(\theta) := g_{t|0}(x_0, \theta_{\times t})$ and $\hat{u}_t(\theta) := \pi_t(\hat{x}_t(\theta), \theta)$. A related concept that is heavily used is the *surrogate cost $F_t$*, which maps a single policy parameter to a real number.

**Definition 2.4** (Surrogate Cost). *The surrogate cost function is defined as $F_t(\theta) := f_t(\hat{x}_t(\theta), \hat{u}_t(\theta))$.*

Figure 1 shows the overall causal structure, from which these concepts follow.

To measure the performance of an online algorithm, we adopt the objective of ***adaptive policy regret***, which has been used by [7, 52]. It is a stronger benchmark than the static policy regret [1, 6] and is more suited to time-varying environments. We use $\{x_t, u_t, \theta_t\}_{t \in \mathcal{T}}$ to denote the trajectory of the online algorithm throughout the paper. The adaptive policy regret $R^A(T)$ is defined as the maximum difference between the cost of the online policy and the cost of the optimal fixed-parameter policy over any sub-interval of the whole horizon $\mathcal{T}$, i.e.,

$$R^A(T) := \max_{I=[t_1:t_2] \subseteq \mathcal{T}} \left( \sum_{t \in I} f_t(x_t, u_t) - \inf_{\theta \in \Theta} \sum_{t \in I} F_t(\theta) \right). \tag{2}$$

In contrast, the (static) policy regret defined in [1, 6] restricts the time interval $I$ to be the whole horizon $\mathcal{T}$. Thus, a bound on adaptive regret is strictly stronger than the same bound on static regret. Adaptive regret is particularly useful in time-varying environments like Examples 2.1 and 2.2 because an online algorithm must adapt quickly to compete against a comparator policy parameter that can change indefinitely with every time interval [32, Section 10.2].

In the general case when surrogate costs $F_{0:T-1}$ are nonconvex, it is difficult (if not impossible) for online algorithms to achieve meaningful guarantees on classic regret metrics like $R^A(T)$ or static policy regret because they do not have oracle optimization solvers or even the exact knowledge of the surrogate costs. Therefore, we introduce the metric of ***local regret***, which bounds the sum of squared gradient norms over the whole horizon:

$$R^L(T) := \sum_{t=0}^{T-1} \|\nabla F_t(\theta_t)\|^2. \tag{3}$$

Similar metrics have been adopted by previous works on online nonconvex optimization [53]. Intuitively, $R^L(T)$ measures how well the online agent chases the (changing) stationary point of the surrogate cost sequence $F_{0:T-1}$. Since the surrogate cost functions are changing over time, the bound on $R^L(T)$ will depend on how much the system $\{g_t, f_t, \pi_t\}_{t \in \mathcal{T}}$ changes over the whole horizon $\mathcal{T}$. We defer the details to Section 3.3.

## 2.2 Contractive Perturbation and Stability

In this section, we introduce two key properties needed for our sub-linear regret guarantees in adaptive online policy selection. We define both with respect to trajectories generated by "slowly" time-varying parameters, which are easier to analyze than arbitrary parameter sequences.

---

[2] $z_t$ is an auxiliary variable to denote the state at $t$ under initial state $x_\tau$ and parameters $\theta_{\tau:t}$.

**Definition 2.5.** *We denote the set of policy parameter sequences with $\varepsilon$-constrained step size by*

$$S_\varepsilon(t_1 : t_2) := \{\theta_{t_1:t_2} \in \Theta^{t_2-t_1+1} \mid \|\theta_{\tau+1} - \theta_\tau\| \leq \varepsilon, \forall \tau \in [t_1 : t_2 - 1]\}.$$

The first property we require is an exponentially decaying, or "contractive", perturbation property of the closed-loop dynamics of the system with the policy class. We now formalize this property.

**Definition 2.6** ($\varepsilon$-Time-varying Contractive Perturbation)**.** *The $\varepsilon$-time-varying contractive perturbation property holds for $R_C > 0, C > 0$, $\rho \in (0, 1)$, and $\varepsilon \geq 0$ if, for any $\theta_{\tau:t-1} \in S_\varepsilon(\tau : t - 1)$,*

$$\left\| g_{t|\tau}(x_\tau, \theta_{\tau:t-1}) - g_{t|\tau}(x'_\tau, \theta_{\tau:t-1}) \right\| \leq C\rho^{t-\tau} \|x_\tau - x'_\tau\|$$

*holds for arbitrary $x_\tau, x'_\tau \in B_n(0, R_C)$ and time steps $\tau \leq t$.*

Intuitively, $\varepsilon$-time-varying contractive perturbation requires two trajectories starting from different states (in a bounded ball) to converge towards each other if they adopt the same slowly time-varying policy parameter sequence. We call the special case of $\varepsilon = 0$ *time-invariant contractive perturbation*, meaning the policy parameter is fixed. Although it may be difficult to verify the time-varying property directly since it allows the policy parameters to change, we show in Lemma 2.8 that time-invariant contractive perturbation implies that the time-varying version also holds for some small $\varepsilon > 0$.

The time-invariant contractive perturbation property is closely related to discrete-time incremental stability [e.g. 45] and contraction theory [e.g. 46], which have been studied in control theory. While some specific policies including DAC and MPC satisfy $\varepsilon$-time-varying contractive perturbation globally in linear systems, in other cases it is hard to verify. Our property is local and thus is easier to establish for broader applications in nonlinear systems (e.g., Example 2.2).

Besides contractive perturbation, another important property we need is the stability of the policy class, which requires $\pi_{0:T-1}$ can stabilize the system starting from the zero state as long as the policy parameter varies slowly. This property is stated formally below:

**Definition 2.7** ($\varepsilon$-Time-varying Stability)**.** *The $\varepsilon$-time-varying stability property holds for $R_S > 0$ and $\varepsilon \geq 0$ if, for any $\theta_{\tau:t-1} \in S_\varepsilon(\tau : t - 1)$, $\left\| g_{t|\tau}(0, \theta_{\tau:t-1}) \right\| \leq R_S$ holds for any time steps $t \geq \tau$.*

Intuitively, $\varepsilon$-time-varying stability guarantees that the policy class $\pi_{0:T-1}$ can achieve stability if the policy parameters $\theta_{0:T-1}$ vary slowly.[3] Similarly to contractive perturbation, one only needs to verify time-invariant stability (i.e., $\varepsilon = 0$ and the policy parameter is fixed) to claim time-varying stability holds for some strictly positive $\varepsilon$ (see Lemma 2.8). The reason we still use the time-varying contractive perturbation and stability in our assumptions is that they hold for $\varepsilon = +\infty$ in some cases, including DAC and MPC with confidence coefficients. Applying Lemma 2.8 for those systems will lead to a small, overly pessimistic $\varepsilon$.

## 2.3 Key Assumptions

We make two assumptions about the online policy selection problem to achieve regret guarantees.

**Assumption 2.1.** *The dynamics $g_{0:T-1}$, policies $\pi_{0:T-1}$, and costs $f_{0:T-1}$ are differentiable at every time step and satisfy that, for any convex compact sets $\mathcal{X} \subseteq \mathbb{R}^n, \mathcal{U} \subseteq \mathcal{R}^m$, one can find Lipschitzness/smoothness constants (can depend on $\mathcal{X}$ and $\mathcal{U}$) such that:*

*1. The dynamics $g_t(x, u)$ is $(L_{g,x}, L_{g,u})$-Lipschitz and $(\ell_{g,x}, \ell_{g,u})$-smooth in $(x, u)$ on $\mathcal{X} \times \mathcal{U}$.*
*2. The policy function $\pi_t(x, \theta)$ is $(L_{\pi,x}, L_{\pi,\theta})$-Lipschitz and $(\ell_{\pi,x}, \ell_{\pi,\theta})$-smooth in $(x, \theta)$ on $\mathcal{X} \times \Theta$.*
*3. The stage cost function $f_t(x, u)$ is $(L_f, L_f)$-Lipschitz and $(\ell_{f,x}, \ell_{f,u})$-smooth in $(x, u)$ on $\mathcal{X} \times \mathcal{U}$.*

Assumption 2.1 is general because we only require the Lipschitzness/smoothness of $g_t$ and $f_t$ to hold for bounded states/actions within $\mathcal{X}$ and $\mathcal{U}$, where the coefficients may depend on $\mathcal{X}$ and $\mathcal{U}$. Similar assumptions are common in the literature of online control/optimization [25, 29, 47].

Our second assumption is on the contractive perturbation and the stability of the closed-loop dynamics induced by a slowly time-varying policy parameter sequence.

**Assumption 2.2.** *Let $\mathcal{G}$ denote the set of all possible dynamics/policy sequences $\{g_t, \pi_t\}_{t \in \mathcal{T}}$ the environment/policy class may provide. For a fixed $\varepsilon \in \mathbb{R}_{\geq 0}$, the $\varepsilon$-time-varying contractive perturbation (Definition 2.6) holds with $(R_C, C, \rho)$ for any sequence in $\mathcal{G}$. The $\varepsilon$-time-varying stability*

---

[3]This property is standard in online control and is satisfied by DAC [1, 3, 6–8] as well as Examples 2.1 & 2.2.

*(Definition 2.7) holds with $R_S < R_C$ for any sequence in $\mathcal{G}$. We assume that the initial state satisfies $\|x_0\| < (R_C - R_S)/C$. Further, we assume that if $\{g, \pi\}$ is the dynamics/policy at an intermediate time step of a sequence in $\mathcal{G}$, then the time-invariant sequence $\{g, \pi\}_{\times T}$ is also in $\mathcal{G}$.*[4]

Note that Assumption 2.2 is on the joint properties of both the dynamical system and the policy class when composed together in a closed loop. The motivation is to generalize two key properties of linear systems under typical reasonable controllers: 1) the effect of past decisions on the current state decays exponentially fast, and 2) if the system is initialized near the origin, it remains near the origin. We generalize these properties via $\varepsilon$-time-varying contractive perturbation (Definition 2.6) and $\varepsilon$-time-varying stability (Definition 2.7) respectively. Although Assumption 2.2 may seem complicated to understand, it is less restrictive than the assumptions in the most closely related work (e.g., [1, 3, 7]) that focus on linear dynamics.

Compared to other settings where contractive perturbation holds globally [1, 8, 54], Assumption 2.2 only requires it to hold locally in a bounded ball $B(0, R_C)$, which becomes important in nonlinear settings. This brings a new challenge because we need to guarantee that the starting state stays within $B(0, R_C)$ whenever we apply this property in the proof. Therefore, in Assumption 2.2, we assume $R_C > R_S + C\|x_0\|$. Similarly, to leverage the Lipschitzness/smoothness property, we require $\mathcal{X} \supseteq B(0, R_x)$ where $R_x \geq C(R_S + C\|x_0\|) + R_S$ and $\mathcal{U} = \{\pi(x, \theta) \mid x \in \mathcal{X}, \theta \in \Theta, \pi \in \mathcal{G}\}$. Since the coefficients in Assumption 2.1 depend on $\mathcal{X}$ and $\mathcal{U}$, we will set $\mathcal{X} = B(0, R_x)$ and $R_x = C(R_S + C\|x_0\|) + R_S$ by default when presenting these constants. The goal is to ensure that the controller never leaves the region where contractive perturbation applies, which is critical for our analysis and again generalizes properties found in the literature (e.g., Examples 2.1, 2.2, and H.1).

For some systems, verifying Assumption 2.2 is straightforward (e.g., Example 2.1). In other cases, we can rely on the following lemma, which can convert a time-invariant version of the property to general time-varying one. We defer its proof to Appendix C.

**Lemma 2.8.** *Suppose Assumption 2.2 holds for $\varepsilon = 0$ and $(R_C, C, \rho, R_S)$, which satisfies $R_C > (C+1)R_S$. Suppose Assumption 2.1 also holds and let $\mathcal{X} := B(0, R_x)$, where $R_x = (C+1)^2 R_S$. Then, Assumption 2.2 also holds for $\hat{\varepsilon} > 0$, $(\hat{R}_C, \hat{C}, \hat{\rho}, \hat{R}_S)$, and $x_0$ that satisfies $(\hat{R}_C - \hat{R}_S)/C$. Here, $\hat{R}_S, \hat{R}_C, \hat{\rho}$ are arbitrary constants that satisfies $R_S < \hat{R}_S < \hat{R}_C < R_C/(C+1)$ and $\rho < \hat{\rho} < 1$. The positive constants $\hat{\varepsilon}$ and $\hat{C}$ are given detailed expressions in Appendix C.*

**Remark 2.9.** *Lemma 2.8 can also be useful when applied to some parameterized controllers for time-invariant nonlinear systems. For example, the well-known "computed torque control" feedback linearization controllers for robotic manipulators (see, e.g., [55]) renders the closed-loop dynamics exponentially stable about an equilibrium, and the feedback gains can be parameterized. Thus, it satisfies Assumption 2.2 in a neighborhood about the equilibrium, via Lemma 2.8. Even with time-invariant dynamics, the time-varying costs (such as tracking a trajectory determined online) provide a setting where selecting the policy parameters online can be beneficial.*

## 3 Method and Theoretical Results

Our algorithm, Gradient-Based Adaptive Policy Selection (GAPS), is inspired by the classic online gradient descent (OGD) algorithm [32, 56], with a novel approach for approximating the gradient of the surrogate stage cost $F_t$. In the context of online optimization, OGD works as follows. At each time $t$, the current stage cost describes how good the learner's current decision $\theta_t$ is. The learner updates its decision by taking a gradient step with respect to this cost. Mapping this intuition to online policy selection, the *ideal* OGD update rule would be the following.

**Definition 3.1** (Ideal OGD Update). *At time step $t$, update $\theta_{t+1} = \prod_{\Theta}(\theta_t - \eta \nabla F_t(\theta_t))$.*

This is because the surrogate cost $F_t$ (Definition 2.4) characterizes how good $\theta_t$ is for time $t$ if we had applied $\theta_t$ from the start, i.e., without the impact of other historical policy parameters $\theta_{0:t-1}$. However, since the complexity of computing $\nabla F_t$ exactly grows proportionally to $t$, the ideal OGD becomes intractable when the horizon $T$ is large.

---

[4]For $\{g, \pi\}_{\times T}$ to be in $\mathcal{G}$, it must satisfy other assumptions about contractive perturbation and stability that we impose on $\mathcal{G}$ but does not need to occur in real problem instances. We only use this assumption in the proof of Theorem 3.6, and it can be made without the loss of generality for time-invariant dynamics and policy classes.

As outlined in Algorithm 1, GAPS uses $G_t$ to approximate $\nabla F_t(\theta_t)$ efficiently. To see this, we compare the decompositions, with key differences highlighted in colored text:

$$\nabla F_t(\theta_t) = \sum_{b=0}^{t} \frac{\partial f_{t|0}}{\partial \theta_{t-b}}\bigg|_{x_0, (\theta_t)_{\times(t+1)}} \quad \text{and} \quad G_t = \sum_{b=0}^{\min\{B-1,t\}} \frac{\partial f_{t|0}}{\partial \theta_{t-b}}\bigg|_{x_0, \theta_{0:t}}. \tag{4}$$

GAPS uses two techniques to efficiently approximate $\nabla F_t(\theta_t)$. First, we *replace the ideal sequence* $(\theta_t)_{\times(t+1)}$ *by the actual sequence* $\theta_{0:t}$. This enables computing gradients along the actual trajectory experienced by the online policy without re-simulating the trajectory under $\theta_t$. Second, we *truncate the whole historical dependence to at most $B$ steps*. This bounds the memory used by GAPS. $\varepsilon$-time-varying contractive perturbation is the key to bound the bias of $G_t$: Intuitively, in the first step, although $\theta_\tau$ becomes more different with $\theta_t$ as $\tau$ decreases, its impact on $f_{t|0}$ decays more quickly (exponentially); In the second step, the terms that we discard are exponentially small with respect to $B$. We provide a rigorous bound of the bias in Theorem 3.2 and a proof outline in Appendix D.

---

**Algorithm 1** Gradient-based Adaptive Policy Selection (GAPS)

---

**Require:** Learning rate $\eta$, buffer length $B$, initial $\theta_0$.
1: **for** $t = 0, \ldots, T-1$ **do**
2:     Observe the current state $x_t$.
3:     Pick the control action $u_t = \pi_t(x_t, \theta_t)$.
4:     Incur the stage cost $c_t = f_t(x_t, u_t)$.
5:     Compute the approximated gradient: $G_t = \sum_{b=0}^{\min\{B-1,t\}} \frac{\partial f_{t|0}}{\partial \theta_{t-b}}\bigg|_{x_0, \theta_{0:t}}$.
6:     Perform the update $\theta_{t+1} = \prod_\Theta(\theta_t - \eta G_t)$.
7: **end for**

---

Algorithm 1 presents GAPS in its simplest form. Although the expression of the partial derivatives contains $\theta_{0:t}$, the time- and space-efficient implementation of GAPS only requires to store $B$ partial derivatives for $B$ previous time steps. Details are given in Algorithm 2 in Appendix B.

Compared to many previous online control algorithms that take a reduction approach based on OCO with Memory, our algorithm can be much more computationally efficient (see Appendix I.4 for an empirical comparison). Specifically, these works [1, 3, 6] take a different *finite-memory reduction* approach toward reducing the online control problem to OCO with Memory [33] by completely removing the dependence on policy parameters before time step $t - B$ for a fixed memory length $B$. In the finite-memory reduction, one must "imaginarily" reset the state at time $t - B$ to be $\mathbf{0}$ and then use the $B$-step truncated multi-step cost function $f_{t|t-B}(\mathbf{0}, \theta_{t-B:t})$ in the OGD with Memory algorithm [1]. When applied to our setting, this is equivalent to replacing $G_t$ in line 1 of Algorithm 1 by $G'_t = \sum_{b=0}^{B-1} \frac{\partial f_{t|t-B}}{\partial \theta_{t-b}} |_{0, (\theta_t)_{\times(B+1)}}$. However, the estimator $G'_t$ has limitations compared with $G_t$ in GAPS. First, computing $G'_t$ requires oracle access to the partial derivatives of the dynamics and cost functions for arbitrary state and actions. Second, even if those are available, $G'_t$ is less computationally efficient than $G_t$ in GAPS, especially when the policy is expensive to execute. Taking MPC (Example 2.1) as an example, computing $G'_t$ at every time step requires solving $B$ MPC optimization problems when re-simulating the system, where $B = \Omega(\log T)$. In contrast, computing $G_t$ in GAPS only requires solving one MPC optimization problem and $O(B)$ matrix multiplications to update the partial derivatives. One may wonder how significant this improvement is, and if it affects the regret. To address this concern, we compare GAPS with Ideal OGD and OGD with Memory in the setting of MPC with confidence coefficients for a 2D double integrator. The simulation results show that GAPS achieve very similar regret with the two benchmarks, while the improvement on computation efficiency is significant (see Appendix I.4).

## 3.1 Bounds on Truncation Error

We now present the first part of our main result, which states that the actual stage cost $f_t(x_t, u_t)$ incurred by GAPS is close to the ideal surrogate cost $F_t(\theta_t)$, and the approximated gradient $G_t$ is close to the ideal gradient $\nabla F_t(\theta_t)$. In other words, GAPS mimics the ideal OGD update (Definition 3.1).

**Theorem 3.2.** *Suppose Assumptions 2.1 and 2.2 hold. Let $\{(x_t, u_t, \theta_t)\}_{t \in \mathcal{T}}$ denote the trajectory of GAPS (Algorithm 1) with buffer size $B$ and learning rate $\eta \leq \Omega((1-\rho)\varepsilon)$. Then, we have*

$$|f_t(x_t, u_t) - F_t(\theta_t)| = O\big((1-\rho)^{-3}\eta\big) \text{ and } \|G_t - \nabla F_t(\theta_t)\| = O\big((1-\rho)^{-5}\eta + (1-\rho)^{-1}\rho^B\big),$$

*where $\Omega(\cdot)$ and $O(\cdot)$ hide the dependence on the Lipschitz/smoothness constants defined in Assumption 2.1 and $C$ in contractive perturbation — see details in Appendix D.1.*

We defer the proof of Theorem 3.2 to Appendix D.1. Note that this result does not require any convexity assumptions on the surrogate cost $F_t$.

## 3.2 Regret Bounds for GAPS: Convex Surrogate Cost

The second part of our main result studies the case when the surrogate cost $F_t$ is a convex function. This assumption is explicitly required or satisfied by the policy classes and dynamical systems in many prior works on online control and online policy selection [1, 3, 6, 54].

The error bounds in Theorem 3.2 can reduce the problem of GAPS' regret bound in control to the problem of OGD's regret bound in online optimization, where the following result is well known: When the surrogate cost functions $F_t$ are convex, the ideal OGD update (Definition 3.1) achieves the regret bound $\sum_{t=0}^{T-1} F_t(\theta_t) - \min_{\theta \in \Theta} \sum_{t=0}^{T-1} F_t(\theta) = O(\sqrt{T})$, when the step size $\eta$ is of the order $1/\sqrt{T}$ [32]. By taking the biases on the stage costs and the gradients into consideration, we derive the adaptive regret bound in Theorem 3.3. Besides the adaptive regret, one can use a similar reduction approach to "transfer" other regret guarantees for OGD in online optimization to GAPS in control. We include the derivation of a dynamic regret bound as an example in Appendix E.

**Theorem 3.3.** *Under the same assumptions as Theorem 3.2, if we additionally assume $F_t$ is convex for every time $t$ and $\mathsf{diam}(\Theta)$ is bounded by a constant $D$, then GAPS achieves adaptive regret*

$$R^A(T) = O\big(\eta^{-1} + (1-\rho)^{-5}\eta T + (1-\rho)^{-1}\rho^B T + (1-\rho)^{-10}\eta^3 T + (1-\rho)^{-2}\rho^{2B}\eta T\big),$$

*where $O(\cdot)$ hides the same constants as in Theorem 3.2 and $D$ — see details in Appendix D.1.*

We discuss how to choose the learning rate and the regret it achieves in the following corollary.

**Corollary 3.4.** *Under the same assumptions as Theorem 3.3, suppose the horizon length $T \gg \frac{1}{1-\rho}$ and the buffer length $B \geq \frac{1}{2}\log(T)/\log(1/\rho)$. If we set $\eta = (1-\rho)^{\frac{5}{2}} T^{-\frac{1}{2}}$, then GAPS achieves adaptive regret $R^A(T) = O((1-\rho)^{-\frac{5}{2}} T^{\frac{1}{2}})$.*

We defer the proof of Theorem 3.3 to Appendix D.1. Compared to the (static) policy regret bounds of [1, 3], our bound is tighter by a factor of $\log T$. The key observation is that the impact of a past policy parameter $\theta_{t-b}$ on the current stage cost $c_t$ decays exponentially with respect to $b$ (see Appendix D for details). In comparison, the reduction-based approach first approximates $c_t$ with $\hat{c}_t$ that depends on $\theta_{t-B+1:t}$, and then applies general OCO with memory results on $\hat{c}_t$ [1, 3]. General OCO with memory cannot distinguish the different magnitudes of the contributions that $\theta_{t-B+1:t}$ make to $\hat{c}_t$, which leads to the regret gap of $B = O(\log T)$.

In the more restrictive setting of linear time-invariant dynamics with the DAC policy class, the results of a concurrent work [10] can also be used to close the $\log T$ gap on static regret of online policy selection. In comparison, Theorem 3.3 considers more general time-varying dynamics and adopts the stronger metric of adaptive regret. As a practical matter, the follow-the-regularized-leader type of algorithm used by [10] is often (much) less computationally efficient than a gradient-based algorithm like GAPS. Nevertheless, [10] made distinct contributions by allowing the state space to be a general Banach space and providing a lower bound for OCO with unbounded memory.

## 3.3 Regret Bounds for GAPS: Nonconvex Surrogate Cost

The third part of our main result studies the case when the surrogate cost $F_t$ is nonconvex. Before presenting the result, we formally define the variation intensity that measures how much the system changes over the whole horizon.

**Definition 3.5** (Variation Intensity). *Let $\{g_t, \pi_t, f_t\}_{t \in \mathcal{T}}$ be a sequence of dynamics/policy/cost functions that the environment provides. The variation intensity $V$ of this sequence is defined as*

$$\sum_{t=1}^{T-1} \sup_{x \in \mathcal{X}, u \in \mathcal{U}} \|g_t(x, u) - g_{t-1}(x, u)\| + \sup_{x \in \mathcal{X}, \theta \in \Theta} \|\pi_t(x, \theta) - \pi_{t-1}(x, \theta)\| + \sup_{x \in \mathcal{X}, u \in \mathcal{U}} |f_t(x, u) - f_{t-1}(x, u)|.$$

Variation intensity is used as a measure of hardness for changing environments in the literature of online optimization that often appear in regret upper bounds (see [57] for an overview). Definition 3.5 generalizes one of the standard definitions to online policy selection. Using this definition, we present our main result for GAPS applied to nonconvex surrogate costs using the metric of local regret (3).

**Theorem 3.6.** *Under the same assumptions as Theorem 3.2, if we additionally assume that $\Theta = \mathbb{R}^d$ for some integer d, then GAPS satisfies local regret*

$$R^L(T) = O\left(\frac{1+V}{(1-\rho)^3\eta} + \frac{\eta T}{(1-\rho)^6} + \frac{\rho^B T}{(1-\rho)^2} + \frac{\eta^3 T}{(1-\rho)^{13}} + \frac{\rho^{2B}\eta T}{(1-\rho)^5}\right),$$

*where $O(\cdot)$ hides the same constants as in Theorem 3.2 — see details in Appendix F.*

We discuss how to choose the learning rate and the regret it achieves in the following corollary.

**Corollary 3.7.** *Under the same assumptions as Theorem 3.6, suppose the horizon length $T \gg \frac{1}{1-\rho}$ and the buffer length $B \geq \frac{1}{2}\log(T)/\log(1/\rho)$. If we set $\eta = (1-\rho)^{\frac{3}{2}}(1+V)^{\frac{1}{2}}T^{-\frac{1}{2}}$, GAPS achieves local regret $R^L(T) = O((1-\rho)^{-\frac{9}{2}}(1+V)^{\frac{1}{2}}T^{\frac{1}{2}})$.*

We defer the proof of Theorem 3.6 to Appendix F. Note that the local regret will be sublinear in $T$ if the variation intensity $V = o(T)$. To derive the local regret guarantee in Theorem 3.6, we address additional challenges compared to the convex case. First, we derive a local regret guarantee for OGD in online nonconvex optimization. We cannot directly apply results from the literature because they do not use ordinary OGD, and it is difficult to apply algorithms like Follow-the-Perturbed-Leader [e.g. 58] to online policy selection due to constraints on information and step size. Then, to transfer the regret bound from online optimization to online policy selection, we show how to convert the measure of variation defined on $F_{0:T-1}$ to our variation intensity $V$ defined on $\{g_t, \pi_t, f_t\}_{t \in \mathcal{T}}$.

A limitation of Theorem 3.6 is that we need to assume $\Theta$ is a whole Euclidean space so that GAPS will not converge to a point at the boundary of $\Theta$ that is not a stationary point. Example 2.2 and Appendix H.3 show that one can re-parameterize the policy class to satisfy this assumption in some cases. Relaxing this assumption is our future work.

## 4 Numerical Experiments

In this section we compare GAPS to strong baseline algorithms in settings based on Examples 2.1 and 2.2. Details are deferred to Appendix I due to space limitations. Appendix I also includes a third experiment comparing GAPS to a bandit-based algorithm for selecting the planning horizon in MPC, and a computation time comparison between GAPS and the alternative gradient approximation of [1].

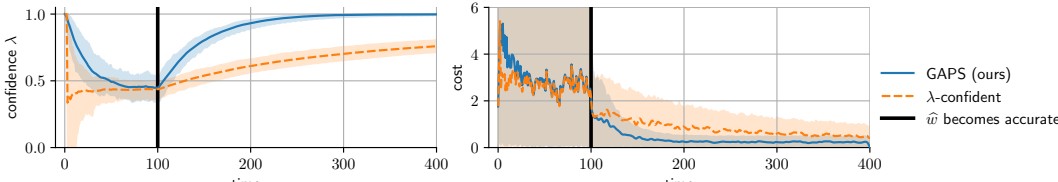

Figure 2: Comparing GAPS and baseline [11] for online adaptation of a confidence parameter for MPC with disturbance predictions. *Left:* Confidence parameter. *Right:* Per-step cost. Shaded bands show 10%-90% quantile range over randomized disturbance properties. See body for details.

**MPC confidence parameter.** We compare GAPS to the follow-the-leader-type method of [11] for tuning a scalar confidence parameter in model-predictive control with noisy disturbance predictions. The setting is close to Example 2.1 but restricted to satisfy the conditions of the theoretical guarantees in [11]. We consider the scalar system $x_{t+1} = 2x_t + u_t + w_t$ under non-stochastic disturbances $w_t$ with the cost $f_t(x_t, u_t) = x_t^2 + u_t^2$. For $t = 0$ to 100, the predictions of $w_t$ are corrupted by a large amount of noise. After $t > 100$, the prediction noise is instantly reduced by a factor of 100. In this setup, an ideal algorithm should learn to decrease confidence level at first to account for the noise, but then increase to $\lambda \approx 1$ when the predictions become accurate.

Figure 2 shows the values of the confidence parameter $\lambda$ and the per-timestep cost generated by each algorithm. Both methods are initialized to $\lambda = 1$. The method of [11] rapidly adjusts to an appropriate confidence level at first, while GAPS adjusts more slowly but eventually reaches the same value. However, when the accuracy changes, GAPS adapts more quickly and obtains lower costs towards the end of the simulation. In other words, we see that GAPS behaves essentially like an instance of Ideal OGD with constant step size, which is consistent with our theoretical results (Theorem 3.2).

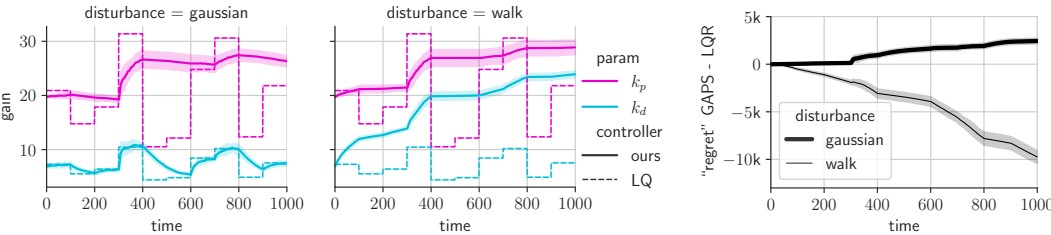

(a) Linear gains $k_p$, $k_d$ tuned by GAPS compared to LQR optimal.

(b) Cumulative cost difference.

Figure 3: Comparing GAPS and LQR baseline in nonlinear inverted pendulum system. Shaded bands show $\pm 1$ standard deviation over the randomness of the disturbances. See body for details.

**Linear controller of nonlinear time-varying system.** We apply GAPS to tune the gain parameters of a linear feedback controller in a nonlinear inverted pendulum system. Every 100 seconds, the pendulum mass changes. The system reflects the smooth nonlinear dynamics and nonconvex surrogate costs in Example 2.2, although it differs in other details (see Appendices H.3 and I.2). We compare GAPS to a strong and dynamic baseline that deploys the infinite-horizon linear-quadratic regulator (LQR) optimal controller for the linearized dynamics at each mass. We simulate two disturbances: 1) i.i.d. Gaussian, and 2) Ornstein-Uhlenbeck random walk.

Figure 3a shows the controller parameters tuned by GAPS, along with the baseline LQR-optimal gains, for each disturbance type. The derivative gain $k_d$ closely follows LQR for i.i.d. disturbances but diverges for random-walk disturbances, where LQR is no longer optimal. This is reflected in the cumulative cost difference between GAPS and LQR, shown in Figure 3b. GAPS nearly matches LQR under i.i.d. disturbances, but significantly outperforms it when the disturbance is a random walk. The results show that GAPS can both 1) adapt to step changes in dynamics on a single trajectory almost as quickly as the comparator that benefits from knowledge of the near-optimal analytic solution, and 2) outperform the comparator in more general settings where the analytic solution no longer applies.

## 5 Conclusion and Future Directions

In this paper, we study the problem of online adaptive policy selection under a general contractive perturbation property. We propose GAPS, which can be implemented more efficiently and with less information than existing algorithms. Under convexity assumptions, we show that GAPS achieves adaptive policy regret of $O(\sqrt{T})$, which closes the $\log T$ gap between online control and OCO left open by previous results. When convexity does not hold, we show that GAPS achieves local regret of $O(\sqrt{(1 + V)T})$, where $V$ is the variation intensity of the time-varying system. This is the first local regret bound on online policy selection attained without any convexity assumptions on the surrogate cost functions. Our numerical simulations demonstrate the effectiveness of GAPS, especially for fast adaptation in time-varying settings.

Our work motivates interesting future research directions. For example, a limitation is that GAPS assumes *all* policy parameters can stabilize the system and satisfy contractive perturbation. A recent work on online policy selection relaxed this assumption by a bandit-based algorithm but requires $\Theta$ to be a finite set [59]. An interesting future direction to study is what regret guarantees can be achieved when $\Theta$ is a continuous parameter set and not all of the candidate policies satisfy these assumptions.

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
