# OpenReview forum: "Online Adaptive Policy Selection in Time-Varying Systems: No-Regret via Contractive Perturbations"
_NeurIPS.cc/2023/Conference — NeurIPS 2023 poster_

### Official Review · Reviewer_5Tn1 · 2023-07-04

**Soundness:** 3 good
**Presentation:** 3 good
**Contribution:** 3 good
**Rating:** 6
**Confidence:** 2

**Summary:**

The paper studies online adaptive policy selection for nonlinear time-varying discrete-time dynamical systems. The algorithm named GAPS is proposed and is shown to achieve optimal regret, which closes the regret gap between online convex optimization (OCO) and online policy selection. En route, a general proof framework based on exponentially decaying perturbation property is developed that connects online policy selection with OCO. Numerical experiments are provided to demonstrate GAPS's superior performance over baselines.

**Strengths:**

The paper is well-written. The problem and the results seem significant.

**Weaknesses:**

A slight weakness is that the algorithm needs $\Omega(\log T)$ memory instead of a constant complexity to $T$. I did not spot any major weakness.

**Questions:**

I hope the authors can provide clarification for two questions:

1.  Can the authors comment on the necessity of assuming Definition 2.6 and Definition 2.7 for achieving optimal regret?

2. Does the variational intensity or a similar quantity also appear in the analysis of Theorem 3.3 (the convex setting)? What is $V$ in that setting?

**Limitations:**

Limitations are discussed in Section 3 and the conclusion. In addition to the ones mentioned in the paper, the algorithm also needs the knowledge of problem parameters $\rho$ and $\epsilon$ (or $\rho$ and $V$) to set the learning rate and run optimally.

---

> ### Author Rebuttal · Authors · 2023-08-10
>
> Thanks for your comments and please find the response to your comments below.
>
> > The algorithm needs $\Omega(\log T)$ memory instead of a constant complexity to $T$.
>
> To the best of our knowledge, there is no algorithm that can achieve sublinear regret with $O(1)$ memory length in our setting. It is possible that this is a fundamental limit of the online policy selection problem.
>
> > The necessity of assuming Definitions 2.6 and 2.7.
>
> We discuss below about the necessity of Definitions 2.6 and 2.7 respectively.
>
> $\varepsilon$-time-varying contractive perturbation (Definition 2.6): This assumption guarantees that the impact of a past decision decays quickly over time. Intuitively, general online policy selection is intractable without an assumption that limits the impact of a bad decision in the past. To see this, consider a setting where the dynamics is $x_{t+1} = x_t, \forall t \geq 2$. It can satisfy all of our assumptions except contractive perturbation. Any online algorithm may suffer a linear regret in this setting because it cannot foresee the future to choose $\theta_1$ optimally before the state `freezes’ at time step $2$.
>
> So we argue that some kind of assumption in the spirit of “forgetting the past” is necessary for sublinear regret in online policy selection. Searching for more general assumptions in this spirit is a future research direction of great interest.
>
> $\varepsilon$-time-varying stability (Definition 2.7): This assumption guarantees that any slowly-time-varying policy parameter sequence can stabilize the system. Without this assumption, it is possible for the trajectory of GAPS to grow to an unbounded magnitude. This will break our approximation error bounds (Theorem 3.2) and regret bounds (Theorems 3.3 and 3.6) because the gradients and parameter updates are no longer uniformly bounded.
>
> > Does the variational intensity or a similar quantity also appear in the analysis of Theorem 3.3?
>
> No. Intuitively, a quantity like variational intensity appears in the regret bound when we allow the comparator policy parameters to change over time. For the metric of adaptive regret in Theorem 3.3, the comparator policy parameter is fixed (see equation (2)). In contrast, for the metric of local regret in Theorem 3.6, one can view any local minimizer of $F_t$ as the comparator policy parameter, which is changing over time. Thus, we need to introduce the variational intensity in the regret bound in Theorem 3.6.
>
> > The algorithm also needs the knowledge of problem parameters $\rho$ and $\varepsilon$ (or $\rho$ and $V$) to set the learning rate and run optimally.
>
> In the literature of online learning/optimization, it is common for the optimal learning rate to depend on the system parameters (see [31] for a survey). In the case when these system parameters are unknown, our Theorems 3.3 and 3.6 also provide regret guarantees for arbitrary learning rates. For example, even when $\rho$ is unknown, one can still achieve $O(\sqrt{T})$ regret in Theorem 3.3 with the learning rate $1/\sqrt{T}$ given that $T >> 1/\varepsilon$.

---

> > ### Comment · Reviewer_5Tn1 · 2023-08-10
> >
> > I appreciate the clarifications from the authors in their rebuttal and maintain my positive rating of this paper.

---

### Official Review · Reviewer_GGAL · 2023-07-07

**Soundness:** 3 good
**Presentation:** 3 good
**Contribution:** 3 good
**Rating:** 6
**Confidence:** 2

**Summary:**

This paper proposes an algorithm, GAPS, for online adaptive policy selection in time-varying systems. The algorithm is shown to achieve optimal $O(\sqrt{T})$ regret based on the contractive perturbation property of the online policy-induced dynamics. Numerical results are provided to verify the performance of GAPS.

**Strengths:**

1. The paper is well-written and organized. The insights behind the main results are effectively presented.
2. The optimal $O(\sqrt{T})$ regret can be achieved using partial derivatives of the dynamics and costs.

**Weaknesses:**

1. The results require a slow change of the policy parameter sequences and an $\epsilon$-time varying contractive perturbation, which cannot handle sudden changes.
2. There is an additional assumption on the initial state.

**Questions:**

1. Is it possible to relax the projection step in the algorithm?
2. It would be interesting to explore the introduction of switching costs in this problem.

**Limitations:**

Yes

---

> ### Author Rebuttal · Authors · 2023-08-10
>
> Thanks for your comments and please find the response to your concerns below.
>
> > The results require a slow change of the policy parameter sequences and an $\varepsilon$-time-varying contractive perturbation.
>
> We require the policy parameter sequences to change slowly for two reasons: 1. Bound the approximation error introduced by using the efficient gradient estimator $G_t$ (see equation (4)). Note that when there is an `abrupt’ change on the policy parameter, our construction of $G_t$ based on the actual trajectory may not approximate $\nabla F_t(\theta_t)$, which comes from using $\theta_t$ repeatedly since time $0$. 2. Satisfy $\varepsilon$-time-varying contractive perturbation (see line 257 in Theorem 3.2). Note that when $\varepsilon = +\infty$ as in the case of Examples 2.1 and H.1, this constraint is always satisfied. And even when $\varepsilon$ is a small constant, the constraint can be easily satisfied because we require $\eta$ to be $O(1/\sqrt{T})$ to achieve the optimal regret (see Corollaries 3.4 and 3.7).
>
> We don’t view our requirement for the policy parameters to change slowly as a limitation because this sequence is fully under the control of our algorithm. GAPS ensures that the parameter $\theta_t$ does not change too fast by using a carefully chosen learning rate $\eta$ (see Theorem D.5 for the formal statement).
>
> > There is an additional assumption on the initial state.
>
> We make this assumption to ensure the critical contractive perturbation property (Definition 2.6) is always satisfied at any state visited by our algorithm. Such assumption is necessary because our $\varepsilon$-time-varying contractive perturbation is locally, and large initial state $x_0$ can make an intermediate state $x_t$ go out of the region $B_n(0, R_C)$ where contractive perturbation holds. This assumption can be relaxed when contractive perturbation holds globally (e.g., Example 2.1).
>
> > Is it possible to relax the projection step in the algorithm?
>
> Thank you for mentioning this point! The projection step is a standard way to handle the constraint in many first-order online optimization algorithms (see [31] for a survey). When the parameter set $\Theta$ is complicated, the projection step might be expensive to compute. An interesting future direction is to design a projection-free algorithm (see Chapter 7 in [31] for a survey) in the setting of online policy selection.
>
> > About introducing the switching costs.
>
> We didn’t see a direct motivation to introduce switching costs on the decisions (which are the policy parameters $\{\theta_t\}$ in our setting), because GAPS already guarantees that the policy parameter sequence changes slowly. However, it would be interesting to consider more general stage costs. For example, we can allow $c_t$ to also depend on some previous states and actions.

---

> > ### Comment · Reviewer_GGAL · 2023-08-15
> >
> > I appreciate the authors for their detailed response. Thanks!

---

### Official Review · Reviewer_r9o3 · 2023-07-18

**Soundness:** 2 fair
**Presentation:** 2 fair
**Contribution:** 2 fair
**Rating:** 5
**Confidence:** 3

**Summary:**

This paper studies online adaptive policy selection for nonlinear time-varying discrete-time dynamical systems. At time step $t \in\mathcal{T}$, the policy picks a control action $u_t$, and the next state and the incurred cost are given by $x_{t+1}=g_t\left(x_t, u_t\right), c_t:=f_t\left(x_t, u_t\right)$, where $g_t(\cdot, \cdot)$ is a time-varying dynamics function and $f_t(\cdot, \cdot)$ is a time-varying stage cost. The goal is to minimize the total cost $\sum_{t=0}^{T-1} c_t$. The regret definition in (2) is similar to [52]( https://arxiv.org/pdf/1708.00075.pdf). They require two key properties to achieve sub-linear regret in this time varying systems. Definition 2.6 requires that Intuitively, two trajectories starting from different states (in a bounded ball) to converge towards each other if they adopt the same slowly time-varying policy parameter sequence, and definition 2.7 requires that the policy class $\pi_{0:T-1}$ can achieve stability if the policy parameters $\theta_{0:T-1}$ vary slowly. Assumption 2.1 is standard (see [52]( https://arxiv.org/pdf/1708.00075.pdf)) while assumption 2.2 ensures starting state stays within an euclidean ball whenever the dynamics changes. Since the complexity of computing $\nabla F_t$ exactly grows proportionally to $t$, the key difference in their approach is that their algorithm GAPS uses $G_t$ to approximate $\nabla F_t\left(\theta_t\right)$ over a batch size of $B$. This results in solving only one MPC optimization problem. Finally in their main theorem 3.6 they show that a regret of $R^L(T)=O\left((1-\rho)^{-\frac{9}{2}}(1+V)^{\frac{1}{2}} T^{\frac{1}{2}}\right)$ is possible without any convexity assumption on $F_t$, where $V$ is the variation intensity of the time-varying system. Finally they empirically validate their algorithm.

**Strengths:**

1) The paper analyzes Online Gradient Descent (OGD) algorithm for time varying systems. They propose the GAPS algorithm which uses the approximate gradient $G_t$ to estimate surrogate function $F_t$. This is also computationally faster than previous methids.

2) They theoretically analyze their algorithm and provide a regret bound of $R^L(T)=O\left((1-\rho)^{-\frac{9}{2}}(1+V)^{\frac{1}{2}} T^{\frac{1}{2}}\right)$ is possible without any convexity assumption on $F_t$, where $V$ is the variation intensity of the time-varying system. This improves over the previous bounds in this setting by a $log(T)$ factor.

3) They show empirically that their algorithm is competitive.

**Weaknesses:**


1) While assumption 2.1 is standard, I think assumption 2.2 is very strong. The implication of the $R_C>R_S+C\left\|x_0\right\|$ in assumption 2.2 is not clear to me. Moreover $\mathcal{G}$ is the set of all possible dynamics/policy sequences ${g_t, \pi_t\}_{t \in \mathcal{T}}$ the environment/policy class may provide and you assume that if $\{g, \pi\}$ is the dynamics/policy at an intermediate time step of a sequence in $\mathcal{G}$, then the time-invariant sequence $\{g, \pi\} \times T$ is also in $\mathcal{G}$. This seems to be very strong assumption. Where do you use it? Are there other works also that require this assumption or is this specific for the time varying system to provide stability?

2) The key novelty in their method lies in using $G_t$ instead of $F_t$ and substituting the ideal sequence by the actual sequence $\theta_{0: t}$. However, doesn't this approach might introduce additional variance in your estimation of the gradient? How do you control for that? Similarly when you truncate your observation to $B$ timesteps rather than the ideal sequence there must be approximation error creeping into your estimation of $F_t$ through $G_t$. How do you account for that? Also it will be great if you can point out where in the theory you deal with these issues.

3) It is not clear to me how the regret improvement occurs in Theorem 3.3 and Theorem 3.6 that results in a regret of $O(\sqrt{T})$ (and improves by a factor of $\log(T)$. The paper has limited discussions on how this happens, and I would like the authors to discuss/clarify this in more details. It will be also great if the authors specifically point out where in their proof the use the assumption 2.1 and 2.2 ti get the improvement. Also why [52]( https://arxiv.org/pdf/1708.00075.pdf) fails to achieve this bound.

4) It makes sense to me that the quantity $V$ occurs in the time-varying system which is similar to the quantity in [Besbes et al.](https://arxiv.org/pdf/1307.5449.pdf). However, it is defined on $f,g$, and policy $\pi$. Shouldn't this $V$ only depend on the environment dynamics and cost $f,g$? Can you please elaborate on this? Also please point out how it comes up in the proof of Theroem 3.6.


**Questions:**

See weakness section.

**Limitations:**

1) The writing can be improved. I think you some of the definitions and assumptions can be moved to the section 3. Also the authors need to discuss the results more. It is not clear to me exactly what technical novelty over [52]( https://arxiv.org/pdf/1708.00075.pdf) led to the regret bound of $O(\sqrt{T})$ which does not include the $\log T$ factor.

2) The two examples in the main paper seems to be slightly contrived (and i did not see the appendix). Can the authors give moire real life examples where their approach can be used?

3) See weakness section.

---

> ### Author Rebuttal · Authors · 2023-08-10
>
> Thanks for your comments. Please see our global response about the generality and complexity of our assumptions. Detailed responses are below.
>
> > The implication of the $R_C > R_S + C\|x_0\|$ in Assumption 2.2 is not clear.
>
> The goal of Assumption 2.2 is to guarantee that the critical contractive perturbation and stability properties (Definitions 2.6 and 2.7) hold on the trajectory of GAPS when its learning rate is small enough. By assuming $R_C > R_S + C\|x_0\|$, we show that any state $x_t$ on the trajectory of GAPS satisfies $\|x_t\| \leq R_S + C\|x_0\|$ (eq. (25) in Appendix D.5), so one can apply contractive perturbation from any intermediate state visited by GAPS to bound the partial derivatives of multi-step dynamics/costs (Lemma D.3, Corollary D.4).
>
> > The assumption that “If $g, \pi$ is the dynamics/policy at an intermediate time-step of a sequence in G, then the time-invariant sequence $g, \pi$ repeating $T$ times is also in G” seems very strong.
>
> To clarify, we only need this repeating sequence of $g, \pi$ to satisfy contractive perturbation and stability, not to occur in real problem instances. We only use this assumption in the proof of Theorem 3.6 (in a rather technical way, see Appendix F). There is no fair comparison with previous works because Theorem 3.6 is the first regret bound for online policy selection with nonconvex surrogate costs. Also, this assumption is without the loss of generality for time-invariant dynamics and policy classes. We will discuss in the revision.
>
> > About variance in our gradient estimation introduced by using the actual parameter sequence $\theta_{0:t}$ and bounded buffer length $B$.
>
> $\varepsilon$-time-varying contractive perturbation (Definition 2.6) is the key property that enables us to bound the bias of our gradient estimation. (Our setting is nonstochastic, so it has no variance.) Intuitively, when the learning rate is small, this property guarantees that the actual state/action pair is close to the state/action pair achieved by applying $\theta_t$ repeatedly since time $0$. This intuition extends to the gradient estimation, formalized in Theorems D.5 and D.6 in the appendix. Discarding the partial derivatives before $b$ steps ago in the expression of $\nabla F_t(\theta_t)$ introduces small bias because the magnitude of this partial derivative decays exponentially with respect to $b$ (see Corollary D.4). We will discuss more about this intuition.
>
> > What enables the regret to improve by a factor of $\log T$ in Theorems 3.3 and 3.6?
>
> As discussed after Corollary 3.4, the regret bounds in [1, 3] are loose by a factor of $B = O(\log T)$ because they apply general OCO with memory results. This treats the impact of all inputs $\theta_{t-B+1:t}$ to the OCO stage cost equally, introducing the factor of $B$. Our problem is more “structured” because the impact of a past parameter $\theta_{t-b}$ on the current $c_t$ decays exponentially with respect to $b$. We formally state this insight in Corollary D.4, which uses Assumptions 2.1 and 2.2.
>
> > Comparison with [52].
>
> [52] studies online nonconvex optimization, so its regret bounds are not directly comparable with our main results for online policy selection. It is also challenging to use our gradient estimator $G_t$ (4) in the algorithms proposed by [52]. This is because $G_t$, constructed using the actual trajectory experienced by GAPS, is only a good approximation of $\nabla F_t(\theta)$ for $\theta$ that is at or very close to $\theta_t$. However, line 7 in Algorithm 1 of [52] queries for the gradients that may be far from $\theta_t$.
>
> > Why should $V$ depend on $f, g,$ and policy $\pi$?
>
> $V$ depends on $\pi_t$ because the online agent can only pick the policy parameters $\theta_t$. The policy classes $\pi_t$ are given. Even under time-invariant costs and dynamics, time-varying policy classes can change how policy parameters affect the states and actions, which lead to the changes in surrogate cost functions $F_t$.
>
> In the proof, the terms that measure the variation on policy classes are introduced in equations (48-50) in the proof of Lemma F.4 in Appendix F.3, where we bound the variation of surrogate cost functions $F_t$ by the variation of $f_t, g_t, \pi_t$.
>
> > The two examples in the main paper seems to be slightly contrived… Can the authors give more real life examples where their approach can be used?
>
> We are not sure if the Reviewer refers to Examples 2.1 and 2.2 or our two numerical experiments.
>
> We contend that Example 2.1 can be broadly useful in settings with a complex, but partially predictable disturbance process. The manuscript’s reference [10] gives examples of EV charging and trajectory tracking. Regarding Example 2.2, it models stabilizing a nonlinear system about an operating point by linearization, a standard textbook technique in control engineering.
>
> Our experiments are meant to clearly demonstrate the properties of GAPS that our theory predicts. Experiment 1 shows the fast adaptation predicted by our adaptive (vs. static) regret bound. Experiment 2 shows that we can handle nonlinear systems. Also, the inverted pendulum is a “wrong but useful” approximation of bipedal walking [Grizzle et al., 2014] and a standard benchmark problem.
>
> For real-life applications, GAPS can be instantiated in almost any system even if one cannot verify all assumptions. The only hard requirement is the smoothness for locally bounded derivatives. By analogy, gradient descent for optimization has strong theoretical guarantees mostly for convex problems, but good empirical performance in much broader settings.
>
> We are excited to extend GAPS to more complex systems in both theory and practice. The examples/experiments in this paper are a starting point, since we focus on theory for now.
>
> [Grizzle et al., 2014] J.W. Grizzle, C. Chevallereau, R.W. Sinnet, A.D. Ames. "Models, feedback control, and open problems of 3D bipedal robotic walking." Automatica (2014).

---

> > ### Comment · Reviewer_r9o3 · 2023-08-16
> > **Response to author's rebuttal**
> >
> > I thank the authors for their response. I have some further questions to understand the paper correctly:
> > - Thank you for clarifying assumption 2.2.
> > - I want to dig deeper into this repeating sequence of $g, \pi$ to satisfy contractive perturbation and stability idea. I understand that real-life problems may not satisfy this and this is only required for proof. However, it is important to know whether this assumption makes the proof trivial or can be removed for future works. Can the authors clarify how this is used in the proof?
> > - Thank you for clarifying the variance of the gradient comment. However, do you have any intuition about how you are controlling the bias?
> > - The writing style of this paper is unsatisfactory. For example, corollary D.4 which actually discusses how we can get the improvement is shifted to the appendix. I think these things should be discussed in the main paper in detail as this paper is more theoretical in nature. I have not checked the proof of corollary D.4 in detail but it is used to prove Theorem D.5 on bounding actual stage cost and Theorem D.6 for bounding the bias.
> > - The buffer length $B$ plays a crucial role in gradient concentration. Can the authors discuss how it is chosen, and how choosing it too small or large affects the proof?
> > - Thank you for your clarification on $V$, and experiments.

---

> > > ### Author Response · Authors · 2023-08-16
> > > **Response to the follow-up questions**
> > >
> > > Thank you for providing valuable feedback on our rebuttal. Please find our response to your follow-up questions below.
> > >
> > >
> > > > How we use the assumption about the repeating sequence of $g, \pi$ in the proof.
> > >
> > >
> > > To show Theorem 3.6, we first show a local regret bound for online nonconvex optimization (Thm F.1) and then use Theorem 3.2 to transfer the regret to online policy selection. In the second step, we need to convert the measure of variation on $F_t$ defined for online optimization (see Thm F.1) to variation intensity $V$ on $g_t, f_t, \pi_t$ defined for control (see Def 3.5). To do the conversion, we adopt an approach that requires the assumption about repeating $g, \pi$, whose insight is discussed below.
> > >
> > >
> > > We realize that $F_t$ is constructed by the sequence of dynamics/policies
> > > $$\pi_0, g_0, \pi_1, g_1, \ldots, \pi_{t-1}, g_{t-1}, \pi_t, f_t,$$
> > > while $F_{t-1}$ is constructed by another sequence of dynamics/policies that is shorter:
> > > $$\pi_0, g_0, \pi_1, g_1, \ldots, \pi_{t-2}, g_{t-2}, \pi_{t-1}, f_{t-1}.$$
> > > Although bounding the distance between $F_t$ and $F_{t-1}$ directly may be challenging, the comparison becomes much easier if we first compare each of them to the auxiliary sequences that repeat $\pi_t, g_t$ for $t$ and $t-1$ times with the help of the assumption (see equation (49)). We can compare repeating $\pi_t, g_t$ with different lengths easily under the assumption because they converge quickly to a limit as shown in Lemma F.3. A formal statement and the detailed proof can be found in Lemma F.4 and Appendix F.3.
> > >
> > >
> > > It is interesting to see if an alternative approach can relax the assumption of repeating $g, \pi$.
> > >
> > >
> > > > Intuition about how to control the bias.
> > >
> > >
> > > The bias on our gradient approximation is controlled by choosing (1) a sufficiently small learning rate $\eta$ and (2) a sufficiently large buffer length $B$. To understand why this works intuitively, we can think about the two sources where the approximation bias comes from.
> > >
> > >
> > > The first source of the bias is that, while we want to evaluate the current policy parameter $\theta_t$, the past policy parameters that lead to the current state are different with $\theta_t$. Under learning rate $\eta$, for a past time step $\tau$, the difference between the parameters can be bound by $\|\theta_t - \theta_\tau\| = O((t-\tau)\eta)$. Thus, under contractive perturbation, the impact of this difference on the current state is $O(\rho^{t-\tau} (t-\tau)\eta)$. The total impact from all previous time steps can be bounded by $O(\eta)$ because the exponentially decaying term $\rho^{t-\tau}$ dominates the linear term $(t-\tau)$. Therefore, we can control this bias by choosing a small learning rate $\eta$.
> > >
> > >
> > > The second source of the bias comes from the truncation using a finite buffer length $B$. Under contractive perturbation, we know $\frac{\partial f_{t\mid 0}}{\partial \theta_\tau} = O(\rho^{t-\tau})$, so the sum of the discarded partial derivative terms under truncation is $O(\rho^B)$. Therefore, we can control this bias by choosing a large buffer length $B$.
> > >
> > >
> > > > Improving the writing style of this paper.
> > >
> > >
> > > Thank you for this valuable comment. From our discussion, we realized that more important theoretical insights could be highlighted in the main body to facilitate understanding. Space is a challenge due to page limits, but we will do our best to move more about the proof outline and add pointers to the appendix.
> > >
> > >
> > > > How the buffer length $B$ affects the proof?
> > >
> > >
> > > The buffer length $B$, either small or large, does not affect our proofs because the bounds in Theorems 3.2, 3.3, and 3.6 take all possible values of $B$ into consideration. However, a small buffer length (e.g., constant) is not sufficient to achieve a sublinear regret, and one can see what regret a specific $B$ can achieve by substituting the value into the bounds in Theorems 3.3 or 3.6. We discuss about the lower bounds of $B$ to achieve the optimal regret bounds in Corollaries 3.4 and 3.7. Note that choosing a larger buffer length $B$ will not make the regret bounds worse.

---

### Official Review · Reviewer_FkR7 · 2023-07-21

**Soundness:** 3 good
**Presentation:** 3 good
**Contribution:** 3 good
**Rating:** 6
**Confidence:** 3

**Summary:**

The paper studies online adaptive policy selection for nonlinear systems. The algorithm proposed by the authors, GAPS, is a gradient-based algorithm that achieves the first optimal regret bound in the convex case, and the first local regret bound in the case when convexity does not hold. The authors provided numerical experiments.

**Strengths:**

The novel approach to the online control problem is interesting and closes the $\log T$ gap between the currently established bounds for online nonstochastic control and OGD. The paper is well-organized.

**Weaknesses:**

1. The experiments do not compare the algorithm proposed by the paper to the existing algorithms in online nonstochastic control. It would be interesting to see the comparison against benchmarks in the online control literature including GPC in Algorithm 1, https://arxiv.org/pdf/2211.09619.pdf.

2. Although the analysis is novel, the algorithm proposed is essentially OGD with approximated gradient. The idea of truncation is also very similar to the gradient-based existing algorithms in online control like GPC.

3. To compute the gradient estimator in GAPS, do we need access to all $\theta_t$'s, requiring storing all $\theta_t$?

**Questions:**

1. Can the authors provide more justifications of Definition 2.7 ($\epsilon$-time-varying stability)? How does this assumption compare with the standard assumptions made in existing literature?

2. One of the main contributions of the paper is that it closes the $\log T$ gap between OCO-M based control algorithms and the OGD regret guarantee. However, there are also OCO-M based algorithm that achieves $O(\sqrt{T})$ bound such as in https://arxiv.org/pdf/2210.09903.pdf. Can the authors compare GAPS to this work?

**Limitations:**

Not applicable.

---

> ### Author Rebuttal · Authors · 2023-08-10
>
> Thanks for your comments and please find the response to your comments below.
>
> > It would be interesting to see the comparison against benchmarks in the online control literature including GPC.
>
> The Gradient Perturbation Control (GPC) in Hazan and Singh, [2022] can be viewed as a special case of our Ideal OGD Update (Definition 3.1) when applied to the disturbance-action controller class in linear time-varying systems. We discussed the major algorithmic differences between GAPS, Ideal OGD, and the finite-memory reduction approach [1] in Section 3.
>
> For the rebuttal, we compared GAPS to the Ideal OGD as well as the gradient approximation of [1]. Plots are shown in the rebuttal supplement. The setting is MPC with confidence coefficients for a 2D double integrator, as discussed in Appendix I.3 of the manuscript. In the computation time plot, we see that the oracle’s computation time grows quadratically and we must terminate it early. GAPS and the method of [1] both use constant time per step, but GAPS’ constant is smaller. On the regret plots, the three methods are indistinguishable. The final regret of GAPS and [1] differ by less than 0.02%, while the computation time of GAPS is over 15x faster. We can include this result in the final paper.
>
> > The idea of truncation is very similar to the gradient-based existing algorithms in online control like GPC.
>
> As we discussed before, GPC can be viewed as a special case of our Ideal OGD Update (Definition 3.1) when applying to the disturbance-action controller class in linear time-varying systems. While Ideal OGD does not use truncation, GAPS and the finite-memory reduction approach [1, 3, 6] use the truncation but they approximate the gradients of the surrogate costs in different ways. As we discussed in line 239, Section 3, the design of GAPS enables it to be implemented much more efficiently with less derivative information than existing approaches.
>
> > Does GAPS require storing all previous parameters $\{\theta_\tau\}_{\tau \leq t}$?
>
> No. The time- and space-efficient implementation of GAPS only requires to store $B$ partial derivatives for $B$ previous time steps (see Algorithm 2 in Appendix B). In revision, we will add a clarification under the simplest form of GAPS (Algorithm 1) that it does not store the previous parameters $\theta_{0:t}$ in the practical implementation.
>
> > Can the authors provide more justifications of Definition 2.7 ($\varepsilon$-time-varying stability)?
>
> Intuitively, $\varepsilon$-time-varying stability holds if any slowly time-varying policy parameter sequence $\theta_{0:T}$ can achieve stability from state $0$. By Lemma 2.8, one only needs to verify this property for $\varepsilon = 0$ (any fixed policy parameter) to claim that this property holds for some strictly positive $\varepsilon$ in our setting. By assuming this property holds, we study the problem of optimizing the policy parameter $\theta_t$ while the stability issue has been handled by the policy class $\pi_t$. Handling the case where not all policy parameters can stabilize the system is challenging, and we leave it as a future direction (see Section 5 for a discussion).
>
> We believe our $\varepsilon$-time-varying stability property is more general than the combination of disturbance-action controller (DAC) class applied to linear systems (e.g., see Section 6.2.4 of Hazan and Singh, [2022]), which is commonly used in many previous works [1, 3, 6]. Specifically, DAC satisfies $\varepsilon$-time-varying stability with $\varepsilon = +\infty$ when applied to linear time-varying systems, which means arbitrary policy parameter sequences can achieve stability (see Appendix H.1). In contrast, our algorithm and theoretical results also apply to settings where $\varepsilon$-time-varying stability only holds for small $\varepsilon$. An example of such settings is linear feedback control in nonlinear systems (see Example 2.2 and Appendix H.3).
>
> > Can the authors compare GAPS to Kumar et al., [2022]?
>
> Thank you for pointing out this related work!  This work does indeed close the $\log T$ regret gap as well, but in a more restricted setting (as discussed below).  Since this paper and the preprint version of our work appeared on arXiv within a week of each other, we will revise our submission to include this paper as concurrent related work.
>
> There are several major differences between Kumar et al., [2022] and our work. For comparison, one shall view the history vector $h_t$ in Kumar et al., [2022] as our state $x_t$, the decision vector $x_t$ as our policy parameter $\theta_t$, where the policy $\pi_t$ is always an identity function (i.e., the control action $u_t = \pi_t(\theta_t) = \theta_t$). Under this mapping of the notations, Kumar et al., [2022] studies a special case of our setting where the dynamics is linear time-invariant and the policy is identity. And one can verify that our Assumptions 1 and 2 hold under their Assumptions A1-A5 when our parameter set $\Theta$ (corresponds to their $\mathcal{X}$) is a convex compact set. The main regret upper bound in Theorem 3.1 of Kumar et al., [2022] should be compared with our Theorem 3.3 and Corollary 3.4 because our surrogate cost $F_t$ is convex in their setting. Both regret bounds are in the order of $O(\sqrt{T})$, while our metric of adaptive regret is stronger than their metric of static regret.  As a practical matter, Kumar et al., [2022] uses a follow-the-regularized-leader type of algorithm, which is often (much) less computationally efficient than our gradient-based algorithm. One distinct contribution of Kumar et al., [2022] is a lower bound for online convex optimization with unbounded memory.
>
> [Hazan and Singh, 2022]: Hazan, Elad, and Karan Singh. "Introduction to online nonstochastic control." arXiv preprint arXiv:2211.09619 (2022).
>
> [Kumar et al., 2022]: Kumar, Raunak, Sarah Dean, and Robert D. Kleinberg. "Online Convex Optimization with Unbounded Memory." arXiv preprint arXiv:2210.09903 (2022).

---

> > ### Comment · Reviewer_FkR7 · 2023-08-15
> > **Thank you for your clarifications.**
> >
> > Thank you for your clarifications. I have no further questions.

---

### Official Review · Reviewer_AQ7q · 2023-07-26

**Soundness:** 3 good
**Presentation:** 3 good
**Contribution:** 3 good
**Rating:** 7
**Confidence:** 2

**Summary:**

This paper proposes a new algorithm, Gradient-based Adaptive Policy Selection (GAPS), for online adaptive policy selection with time-varying dynamics and costs. For analysis, it proposes a general analytical framework for online policy selection. Under this framework, by restricting the problem and policy class to have the contractive perturbation property it identified, GAPS is shown to approximate an ideal online gradient descent algorithm. This results in better regret bounds compared to existing results. When convexity holds, GAPS is the first to achieve optimal regret in this setting; when convexity doesn’t hold, it gives the first local regret bound for online policy selection. Empirical results on two examples in the main text also illustrate GAPS’s better adaptivity to changing environments compared to baselines.

**Strengths:**

1. The results seem to be significant. First, when assuming convexity, the regret bound of the proposed algorithm improves over existing work and fills a gap in the literature. It is the first to achieve the optimal regret of $O(\sqrt{(T)})$ under the discussed setting while requiring less information about the problem. Second, when the cost function is nonconvex, it gives the first local regret bound for online policy selection.
2. The proposed contractive perturbation property and its corresponding analytical framework are general and subsume an existing class (DAC) as well as some known downstream applications. They may be helpful for future research.
3. The paper is well-written, and the presentation of empirical results is clear, which is convincing.

**Weaknesses:**

Overall, the paper seems strong to me, and I have a minor suggestion: If possible, it would be nice to have the result of online gradient descent (OGD) oracle in the numerical experiments. This may help the reader gain more understanding about GAPS’s performance.

**Questions:**

1. Though the proposed contractive perturbation property includes some existing works as special cases, I wonder how likely or hard it is to find a more general property.

Typos
1. In Line 216, it seems that some part is missing after “satisfies”.

**Limitations:**

As the paper discussed in the Conclusion and Future Directions section, the major limitation of this work may be that the assumptions on the contractive perturbation property and stability are quite strong. It requires the properties should hold for all policy parameters. But still, this work is quite complete, and relaxing the assumptions can be interesting future directions.

---

> ### Author Rebuttal · Authors · 2023-08-10
>
> Thanks for your comments and please find the response to your concerns below.
>
> > It would be nice to have the result of online gradient descent (OGD) oracle in the numerical experiments.
>
> Thank you for the suggestion. For the rebuttal, we performed an experiment comparing GAPS to the OGD oracle as well as the gradient approximation from the manuscript’s reference [1], which we discussed in lines 239 of the manuscript. Plots are shown in the rebuttal supplement. The setting is MPC with confidence coefficients for a 2D double integrator, as discussed in Appendix I.3 of the manuscript. In the computation time plot, we see that the oracle’s computation time grows quadratically and we must terminate it early. GAPS and the method of [1] both use constant time per step, but GAPS’s constant is smaller. On the regret plots, the three methods are indistinguishable. The final regret of GAPS and [1] differ by less than 0.02%, while the computation time of GAPS is over 15x faster. We can include this result in the final paper.
>
> > I wonder how likely or hard it is to find a more general property than contractive perturbation.
>
> The goal of contractive perturbation is to guarantee the impact of a previous decision decays quickly over time as long as the policy parameter is changing sufficiently slowly. We believe a similar decay property is needed to address the challenge of indefinite impact of a past error in policy selection. An interesting direction towards relaxing it is to consider the case where only a subset of policy parameters satisfies this property. We will add a discussion about this intuition in revision.
>
> > The assumptions on the contractive perturbation property and stability are quite strong. It requires the properties should hold for all policy parameters.
>
> Thanks for pointing this out! As we discussed in Section 5, an interesting future direction is to study what guarantees can be achieved when not all of the candidate policy parameters satisfy these assumptions. It is challenging to detect and rule out the policy parameters that violate these properties when $\Theta$ is a continuous parameter set, so we leave this direction as future work.

---

> > ### Comment · Reviewer_AQ7q · 2023-08-15
> >
> > Thank you for the new experiment and your explanation. The additional results look promising and answer my question about the experiments. I don't have other questions and maintain my assessment.

---

### Official Review · Reviewer_hsz4 · 2023-07-26

**Soundness:** 3 good
**Presentation:** 2 fair
**Contribution:** 3 good
**Rating:** 5
**Confidence:** 1

**Summary:**

The paper studies online adaptive policy selection in systems with time-varying costs and dynamics.

This paper proposes an algorithm that obtains optimal regret bound in the convex case and a local regret bound in the non-convex setting with four assumptions: (1) the dynamics are contractive starting from a ball near 0 if the policy has small variations across time. (2) the dynamics starting from 0 never goes out of an even smaller ball if the policy has small variations across time. (3) the dynamics will start from a point that would never go out of the ball in assumption (1). (4) smoothness and Lipschitzness for the dynamics, policy function, and cost functions.

The algorithm does not require Oracle access to the dynamics.

**Strengths:**

The problems under investigation are interesting.

**Weaknesses:**

It seems that the restriction on the dynamics is quite severe.

**Questions:**

Could you give some examples of dynamics where the assumptions hold? I am giving a low score mainly because I am not very sure that I understand the dynamics of interest. If there are good examples which show that the assumptions are not as severe as vacuous, then I will change my score.

---

> ### Author Rebuttal · Authors · 2023-08-10
>
> Thanks for your comments and please find the response to your concerns below.
>
> Our assumptions generalize the assumptions in the most closely related previous work on online control – please see the global rebuttal for details.
>
> The intricate nature of the contractiveness and stability ball assumptions come from our desire to have local, instead of global, assumptions. Note that in our DAC and MPC examples (Appendix H.1 and H.2) the contractiveness ball has radius $\infty$.
>
> The example settings discussed in our paper have each appeared in related work previously with motivating applications:
>
> - Example 1: Learning-augmented model predictive control in linear time-varying systems (Example 2.1), which generalizes the setting studied in the previous work [10] on learning-augmented control. This setting has applications in EV charging and trajectory tracking [10], and we show that it satisfies all of our assumptions in Appendix H.2.
>
> - Example 2: Linear feedback control applied to time-varying nonlinear control (Example 2.2), which was studied in [11, 46] on nonlinear control. This example is practical because it is a standard technique in control engineering to stabilize a nonlinear system about an operating point by linearizing the system and using linear control synthesis. We show it satisfies all of our assumptions in Appendix H.3.
>
> - Example 3: Disturbance-action controllers in linear time-varying systems (Appendix H.1), which has received much attention from previous works on no-regret online control [1, 3, 6]. We show it satisfies all of our assumptions in Appendix H.1.
>
> More discussion: The literature on nonlinear control contains many examples of a parameterized family of controllers for a time-invariant system, each of which renders the closed-loop dynamics exponentially stable about an equilibrium. For example, the well-known “computed torque control” feedback linearization controllers for robotic manipulators, where the feedback gains can be parameterized. These settings satisfy our assumptions in a neighborhood about the equilibrium, via our Lemma 2.8.  Even with time-invariant dynamics, the time-varying costs (such as tracking a trajectory determined online) provide an online, possibly adversarial, setting where our algorithm is useful.

---

> > ### Comment · Reviewer_hsz4 · 2023-08-18
> >
> > Thanks for your classification.

---

> > > ### Author Response · Authors · 2023-08-18
> > >
> > > Thank you for reading and responding to our rebuttal. Please let us know if you have any further questions.

---

### Official Review · Reviewer_1Vdt · 2023-08-02

**Soundness:** 3 good
**Presentation:** 4 excellent
**Contribution:** 3 good
**Rating:** 6
**Confidence:** 2

**Summary:**

This paper presents a novel algorithm, Gradient-based Adaptive Policy Selection (GAPS), for online policy selection in time-varying systems. The authors introduce a general analytical framework for online policy selection via online optimization. The paper also provides theoretical guarantees for the performance of the GAPS algorithm under some assumptions on the stability of the dynamical system and on the convexity of the surrogate cost function. Complementary local bounds are also give in the case of nonconvex cost.

**Strengths:**

The paper is well-organized and easy to follow, with clear explanations of the theoretical concepts and practical implementation details. The authors provide detailed proofs of their theoretical results in the appendices, as well as numerical experiments to document the performance of the GAPS algorithm in two concrete example settings.

On the math side, while I did not check the details of the proof, I find the underlying perturbative idea new and interesting, and on a high level the steps of the proof check out.

**Weaknesses:**

The main weakness of the paper is, given its novelty, that it is hard for the reader to understand how restrictive the set of assumptions that are put on the dynamical systrem. This holds for both assumptions of Theorem 3.3: Convexity of $F$ and $\epsilon$-time varying contractive perturbation/stability. It is good that the authors give examples of systems where these assumptions hold, and describe in Lemma 2.8 how the time-invariant stability can be translated to these conditions, but I still find it a bit hard to understand how restrictive these conditions are. For instance, I imagine that in the case of a multistable dynamical system the contractive perturbation property would not hold?

**Questions:**

one main question listed in the weaknesses.

**Limitations:**

Some of the limitations havebeen listed in the conclusions, but perhaps a more extended discussion about the applicability of the assumptions would be informative.

---

> ### Author Rebuttal · Authors · 2023-08-10
>
> Thanks for your comments.
>
> Our major assumption (Assumption 2.2) is about the joint properties of both the dynamical system and the policy class when composed together in a closed loop. Thus, it is not particularly restrictive on the dynamical system when one has the freedom to choose/design the corresponding policy class. An example is the design of the disturbance-action controller (DAC) class for linear time-varying (LTV) systems (see Appendix H.1), which is a special case of our setting and has been studied by many previous works in online control [1, 3, 6-8]. Assumption 2.2 is also satisfied by other online control settings including learning-augmented model predictive control (MPC) for LTV systems [10] and linear feedback control in nonlinear systems [11, 46] (see Examples 2.1 and 2.2). We will add a discussion about this in the revision.
>
> The literature on nonlinear control contains many examples of a parameterized family of controllers for a time-invariant system, each of which renders the closed-loop dynamics exponentially stable about an equilibrium. For example, the well-known “computed torque control” feedback linearization controllers for robotic manipulators, where the feedback gains can be parameterized. These settings satisfy our assumptions in a neighborhood about the equilibrium, via our Lemma 2.8. Even with time-invariant dynamics, the time-varying costs (such as tracking a trajectory determined online) provide an online, possibly adversarial, setting where our algorithm is useful.
>
> We also want to emphasize that our result about the local regret of GAPS (Theorem 3.6) does not require the surrogate cost $F_t$ to be convex. The convexity assumption is required or satisfied by many previous works on online control [1, 3, 6, 53], and relaxing it is one of our major contributions.
>
> Lastly, a dynamical system with multiple stable equilibrium points can still satisfy the contractive perturbation property (Definition 2.6) because the property is only assumed locally in the ball $B_n(0, R_C)$. By controlling the step size of GAPS (Algorithm 1), we can guarantee that the state of GAPS always stays within $B_n(0, R_C)$ (see (25) in Appendix D.5). Therefore, it does not matter if there are other stable equilibrium points out of $B_n(0, R_C)$.

---

> > ### Comment · Reviewer_1Vdt · 2023-08-16
> > **Response to rebuttal**
> >
> > I thank the authors for their response. My positive assessment remains unchanged.

---

### Author Rebuttal · Authors · 2023-08-10

Several reviewers were concerned that our assumptions are restrictive. We argue that they are perhaps more complicated, but *less* restrictive than the assumptions in the most closely related work (e.g., [1, 3, 7]). In particular, we relax the common linear-time-varying dynamics assumption, and our assumptions are local, which becomes important in nonlinear settings.

The motivation of our assumptions is to generalize two key properties of linear systems under typical controllers: 1) the effect of past decisions on the current state decays exponentially fast, and 2) if initialized near the origin, they remain near the origin. These are generalized by contractive perturbation (Definition 2.6) and time-varying stability (Definition 2.7) respectively.

The remaining assumptions are more technical in nature, but their main purpose is to ensure that the online controller 1) never leaves the region where contractive perturbation applies, and 2) the magnitude of the state does not grow to be unbounded. These two properties are critical for our analysis of GAPS, and are again generalizations of properties found in the literature (e.g., Examples 2.1, 2.2, and H.1).

Please see our responses to the individual reviews for more details about the generality/strength of our assumptions.

---

### Decision · Program_Chairs · 2023-09-21

**Decision:**

Accept (poster)

**Comment:**

This paper had generally favorable reviews along with one strong support in the form a score of 7. The reviewers appreciated the presentation and the non-convex local regret bound. On the other hand, I hope the authors can, in accordance with the reviewers' wishes, expand on the discussion regarding the applicability of the contractivity assumption to nonlinear systems.

I am happy to tentatively recommend this paper for acceptance.